# TEXT INFILLING

## ABSTRACT

Recent years have seen remarkable progress of text generation in different contexts, including the most common setting of generating text from scratch, the increasingly popular paradigm of retrieval and editing, and others. Text infilling, which fills missing text portions of a sentence or paragraph, is also of numerous use in real life. Previous work has focused on restricted settings, by either assuming single word per missing portion, or limiting to single missing portion to the end of text. This paper studies the general task of text infilling, where the input text can have an arbitrary number of portions to be filled, each of which may require an arbitrary unknown number of tokens. We develop a self-attention model with segment-aware position encoding for precise global context modeling. We further create a variety of supervised data by masking out text in different domains with varying missing ratios and mask strategies. Extensive experiments show the proposed model performs significantly better than other methods, and generates meaningful text patches.

## 1 INTRODUCTION

Text generation spans a rich set of tasks that aim to generate natural language from input data. Popular tasks include machine translation, summarization, dialogue, and others. Previous work has made remarkable progress in text generation in various contexts. For example, the most common setting is to generate an entire text sequence from scratch (Mikolov et al., 2010; Sutskever et al., 2014; Bahdanau et al., 2014). Recent work additionally leverages retrieved reference text to help with generation (Guu et al., 2017; Weston et al., 2018), and others (Hu et al., 2017; Shen et al., 2017) generate by manipulating specific aspects of given text.

Text infilling, which fills missing text snippets of a sentence or paragraph, is also a common application in real life useful in numerous contexts, such as restoration of historical or damaged documents, contract or article writing with templates, text editing, and so forth. The counterpart application in visual domain is *image inpainting* (filling missing pixels in images) which has attracted great research and industrial interest and achieved impressive results (Bertalmio et al., 2000; Criminisi et al., 2004; Liu et al., 2018; Yu et al., 2018b). Text infilling, in contrast, is less explored or has been studied in simplified and more restricted settings. For example, the recent MaskGAN work (Fedus et al., 2018) and the sentence completion task (Zweig & Burges, 2011) have assumed each missing portion of a sentence contains only a *single* word. The assumption fails to meet the general text infilling need that each part can miss an arbitrary number of tokens and the missing word count is unknown *a priori*. Other work (Holtzman et al., 2018; Fan et al., 2018) assume the missing text are at the end of a sentence or paragraph, and continuations of the given text are generated. Sun et al. (2017) study image captioning with a single blank surrounded by known text. These studies are not directly applicable to many real scenarios where multiple portions at random positions of the text can be missing.

In this paper, we study the general task of text infilling. Consider input text where an arbitrary number of portions are missing and each portion may originally contain an arbitrary unknown number of tokens. The task aims to infill the missing portions based on the global and surrounding context, to make the text complete and meaningful. For example, given an incomplete sentence (which we call a *template*) "*this is one ____ and i ____ food here*", the desired output could be "*this is one of my favorite restaurants and i really like the food here*". To the best of our knowledge, such general, unconstrained text infilling setting has not been studied previously. We created a variety of supervised data by masking out snippets of sentences with different strategies.

We explore solutions to the task, such as the common attentional sequence-to-sequence model (Bahdanau et al., 2014) and GAN-based approach (Goodfellow et al., 2014). In particular, to better capture the global and surrounding context of the missing portions, we leverage a multi-head self-attention model (Vaswani et al., 2017), and devise a segment-aware position encoding mechanism to enable precise localization when there are multiple missing segments and varying number of missing tokens in each.

We conduct extensive experiments in multiple concrete setups, including randomly masked text of varying number of segments and missing ratios. We also study infilling more specific content, such as prepositions and verbs, as well as infilling large portions based on a few anchor words. Results show our proposed model captures semantic meanings and generates patches that fit into the text.

## 2 RELATED WORK

**Text Generation**   Neural sequence models such as recurrent network (Mikolov et al., 2010) are commonly used in text generation. *Attention mechanism* allows the model to search for a set of positions in the source sentence where the most relevant information is concentrated during decoding. Vaswani et al. (2017) propose the *Transformer*, a model architecture that relies entirely on an attention mechanism (Bahdanau et al., 2014) to draw global dependencies between input and output. New approaches related to deep generative models such as VAE (Kingma & Welling, 2013) and GAN (Goodfellow et al., 2014) have drawn significant attention and give a promising performance in vision domain. When it comes to text generation, Yu et al. (2017) introduce *GAN* and use policy gradient to train the generative model. Bowman et al. (2015) combine the architecture of a variational autoencoder with the RNN-based language model to explicitly capture global features in a continuous latent variable. Hu et al. (2017) combines VAEs and holistic attribute discriminators for an effective imposition of semantic structures. Generative language models with recursive network structure are usually trained with maximum likelihood in an approach known as *teacher forcing*. While such approach may reproduce language patterns in training data, its performance is not as promising when sampling on sequences that were never conditioned on during training. To reduce the impact of this problem, Bowman et al. (2015) introduce an RNN-based variational autoencoder generative model that incorporates distributed latent representations of entire sentences. The result is promising when the VAE model is trained with data imputed with missing words and then tested on sentences whose final 20% are imputed. Fedus et al. (2018) introduce an actor-critic conditional GAN that fills in missing text conditioned on the surrounding context to reduce the GAN's mode dropping problem in the textual setting.

**Image Inpainting**   Filling missing pixels in an image, often referred to as image inpainting or image completion, is a task that has attracted much attention (Barnes et al., 2009; Pathak et al., 2016; Iizuka et al., 2017; Yang et al., 2017; Liu et al., 2018; Yan et al., 2018; Yu et al., 2018a). Current approaches to tackle the hole-filling problem can be divided into two groups. The first set of methods make use of large external databases and attempt to fill in the hole by searching for a patch with similar surroundings (Hays & Efros, 2007). The other set of methods synthesize the texture with different techniques. Barnes et al. (2009) propose *PatchMatch* to quickly find approximate nearest neighbor matches between image patches. Pathak et al. (2016) propose *Context Encoder* to directly predict the missing region with an encoder-decoder CNN. To train this image completion network to be consistent both locally and globally, Iizuka et al. (2017) use global and local context discriminators that are trained to distinguish real images from completed ones. To create a filling patch with finely detailed texture, Yan et al. (2018) introduce a special shift-connection layer to the U-Net architecture while Yang et al. (2017) propose a multi-scale neural patch synthesis approach based on joint optimization of image content and texture constraints.

## 3 TEXT INFILLING

### 3.1 PROBLEM DEFINITION

In this paper, we consider the problem of text infilling: Given a text template where portions of a body of text are deleted or redacted, we want to fill in the blanks in a way that fit into the global semantic structure and provide meaningful details.

**Original Text :**   I want some lobster served with cheese , please .

**Template :**   I want some ___m___ , please .

Figure 1: An example of template creation.

**Data**   In our infilling task setting, a sentence may be viewed as the joint of a set of segments. Each segment consists of multiple tokens in a row. By masking out certain segments(deterministically or stochastically), we receive a template for the input sentence.

Let ___m___ be a placeholder for a blank, where multiple tokens in a row are masked out. In Figure 1, the original text is split into three segments as "I want some", "lobster served with cheese" and ", please .". In the template, the second segment is masked out and replaced by the special token ___m___.

**Notations**   For better illustration, we introduce the following notations:

For the input sequence $\boldsymbol{x} = (\boldsymbol{s}_0, \boldsymbol{s}_1, ..., \boldsymbol{s}_n)$, $\boldsymbol{s}_i$ refers to the $i_{th}$ input segment. Let $x_{(i,j)}$ denote the $j_{th}$ token in the $i_{th}$ input segment $\boldsymbol{s}_i$, $\boldsymbol{s}_i$ can be represent as $(x_{(i,0)}, x_{(i,1)}, ..., x_{(i,o_i)})$. The input sequence may also be given as $\boldsymbol{x} = (x_{(0,0)}, x_{(0,1)}, ..., x_{(0,o_0)}, x_{(1,0)}, x_{(1,1)}, ..., x_{(1,o_1)}, ..., x_{(n,0)}, x_{(n,1)}, ..., x_{(n,o_n)})$.

Let $\boldsymbol{x}_{template_i}$ denote the template sequence that is attended to fill in the blank whose $seg\_id$ is $i$. We use $\boldsymbol{s}'_i$ to refer to the filled-in segment for the blank with $seg\_id = i$ while $x'_{(i,j)}$ denotes a token in it. Finally, let $\mathbb{M}$ be the set that contains all the blanks' $seg\_id$.

## 3.2 APPROACH

Figure 2 depicts the overall architecture of our model. The basis for our model is a multi-head self-attention token decoder, which fits the task of infilling as it is able to condition on information from both the past and the future. Our implementation replicates Vaswani et al. (2017).

### 3.2.1 TEMPLATE

**Initialize Template**   For a discrete input sentence with $n$ segments $\boldsymbol{x} = (\boldsymbol{s}_0, \boldsymbol{s}_1, ..., \boldsymbol{s}_n)$, we first generate a template by masking out part of the token segments deterministically or stochastically. The blanks will be replaced by a special token ___m___ in the template.

For instance, if we mask out segment $i$ and segment $j$ in $\boldsymbol{x}$ $(i < j)$, then $\mathbb{M} = \{i, j\}$. The initial template is given as

$$\boldsymbol{x}_{template} = \boldsymbol{x}_{template_i} = (\boldsymbol{s}_0, ..., \boldsymbol{s}_{i-1}, \_\_m\_\_, \boldsymbol{s}_{i+1}, ..., \boldsymbol{s}_{j-1}, \_\_m\_\_, \boldsymbol{s}_{j+1}, ..., \boldsymbol{s}_n) \tag{1}$$

**Update Template**   After filling in each blank, we update the template by replacing the specific placeholder ___m___ into corresponding segment. During training, after generating the $i_{th}$ segment, the ground truth $\boldsymbol{s}_i$ will be filled back into the template, and $\boldsymbol{x}_{template_i}$ in equation 1 will be updated into $\boldsymbol{x}_{template_j} = (\boldsymbol{s}_0, ..., \boldsymbol{s}_{i-1}, \boldsymbol{s}_i, \boldsymbol{s}_{i+1}, ..., \boldsymbol{s}_{j-1}, \_\_m\_\_, \boldsymbol{s}_{j+1}, ..., \boldsymbol{s}_n)$. During testing, the inference segment $\boldsymbol{s}'_i$ will be filled back into the template, and the new template will be $\boldsymbol{x}_{template_j} = (\boldsymbol{s}_0, ..., \boldsymbol{s}_{i-1}, \boldsymbol{s}'_i, \boldsymbol{s}_{i+1}, ..., \boldsymbol{s}_{j-1}, \_\_m\_\_, \boldsymbol{s}_{j+1}, ..., \boldsymbol{s}_n)$. The decoder will attend to the updated template $\boldsymbol{x}_{template_j}$ when filling in next blank, whose $seg\_id$ is $j$.

### 3.2.2 POSITION ENCODING

Since the Self-attn architecture based solely on attention mechanism and thus contains no recurrence or convolution, we need to inject additional information about the relative or absolute position of the tokens in the sequence.

As can be seen in Figure 2, the location of each token in the template can be uniquely determined by its segment number *seg_id* and the offset in that segment, which we denote as *offset_id*. As in

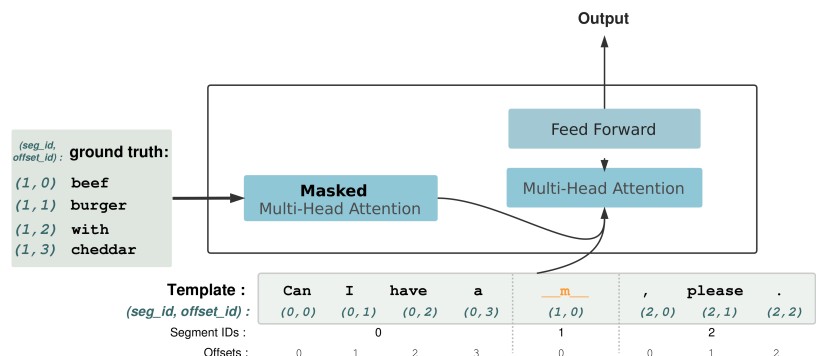

Figure 2: The overall structure of Self-attn. This figure depicts the training process. The decoder will attend to the template at each position, conditioning on the template together with what has been filled in the template. During inference, the input will not go through the masked multi-head attention layer.

original Transformer Vaswani et al. (2017), we use sine and cosine functions of different frequencies as positional embedding:

$$PE_{(pos,2i)} = sin(pos/10000^{2i/d_{model}})$$
$$PE_{(pos,2i+1)} = cos(pos/10000^{2i/d_{model}})$$

where $i$ is the dimension and $pos = seg\_id * base + offset\_id$ is the unique position index for each token given by $(seg\_id, offset\_id)$ and a self-defined integer $base$.

The positional embeddings have the same dimension $d_{model}$ as the word embeddings, ensuring that the two can be summed. The sum of the positional embeddings and the word embeddings for the input token sequence will be used as input for the Transformer.

### 3.2.3 APPLICATIONS OF ATTENTION

As proposed by Vaswani et al. (2017), an attention function maps a query and a set of key-value pairs to an output, where the query, keys, values, and output are all vectors. The input consists of queries and keys of dimension $d_k$, and values of dimension $d_v$. We pack a set of queries, keys and values into matrix $\boldsymbol{Q}$, $\boldsymbol{K}$ and $\boldsymbol{V}$ representatively to compute the attention function simultaneously. The attention function is given by:

$$Attention(\boldsymbol{Q}, \boldsymbol{K}, \boldsymbol{V}) = softmax(\frac{\boldsymbol{Q}\boldsymbol{K}^T}{\sqrt{d_k}})$$

Multi-head attention mechanism projects queries, keys and value to different representation subspaces and calculates corresponding attention. The attention function outputs are concatenated and projected again before giving the final output. Multi-head attention allows the model to attend to multiple features at different positions.

In this work, the multi-head attention is used in the following two ways: (1) The decoder contains self-attention layers where the keys, values and queries come from the output of the previous layer in the decoder. This allows the decoder to attend to all previous positions and make use of local information during infilling. (2) In "template-decoder attention" layers, the queries come from the previous decoder layer, and the template embeddings are used as memory keys and values. This makes sure the decoder can attend to all positions in the template and capture global semantic information while filling each blank.

### 3.2.4 TRAINING

**Objective** In the infilling process, the decoder will fill in the blanks one by one. For the infilling of each segment, the decoder fills in the missing token auto-regressively, conditioning on the template together with what has been filled in the template. To fill the blank with $seg\_id = i$, the objective is

to minimized the following cross-entropy loss:

$$\mathcal{L}_i(x'_{(i,0)}, x'_{(i,1)}, ..., x'_{(i,o_i)} | \boldsymbol{x}_{template_i}) = -\log \prod_{j=0}^{o_i} P(x'_{(i,j)} | x'_{(i,0)}, ..., x'_{(i,j-1)}, \boldsymbol{x}_{template_i}), i \in \mathbb{M}.$$

The loss $\mathcal{L}$ for each infilling sentence is the sum of the cross-entropy loss for each infilling blank: $\mathcal{L} = \sum \mathcal{L}_i, i \in \mathbb{M}$.

**Optimizing** We use Adam optimizer ((Kingma & Ba, 2014)) with $\beta_1 = 0.9$, $\beta_2 = 0.997$ and $\epsilon = 10^{-9}$. We follow the setting in Vaswani et al. (2017) and linearly increase the learning_rate for the first *warmup_steps* training steps, then decrease the learning_rate proportionally to the inverse square root of the step number. We set *const* = 0.3 and *warmup_step* = 10000.

$$learning\_rate = const * \frac{1}{\sqrt{d_{model}}} * min(\frac{step\_num}{(\sqrt{warmup\_step})^3}, \frac{1}{\sqrt{step\_num}})$$

## 4 EXPERIMENTS

In this section, we evaluate the effectiveness of the multi-head self-attention model for the infilling task in various settings. For all the experiments, we compare our model (Self-attn) to two baselines: attentional sequence-to-sequence model (Seq2Seq) (Bahdanau et al., 2014) and GAN-based approach (GAN) (Goodfellow et al., 2014). Aside from the recurrent neural network's ability to capture relative position, we inject positional embedding (section 3.2.2) to Seq2Seq and GAN baseline models to keep the settings the same.

### 4.1 VARYING MASK RATES AND SEGMENTS

As a start, we conduct a set of tests to find the impact template structure has on infilling performance. We test the performance of different generative models when removing different portions of the tokens. With the same mask rate, we generate templates with a different number of blanks.

**Dataset** We use the Yelp text corpus to train the generative models. This dataset consists of 450k food reviews taken from Yelp. We select positive reviews that contain at most 16 words. The resulting dataset contains 104k sentences for training and 1k sentences for testing with the vocabulary size of 9k. Token segments are masked out randomly in the template.

**Quantitative and Human Evaluation** We use multiple metrics to evaluate the quality of the infilling sentence generated by our model(Self-attn), Seq2Seq and GAN. For quantitative evaluation, we compare the perplexity and the BLEU score Papineni et al. (2002) on the test set.

We also invite ten university students to conduct an unbiased human evaluation. Under each setting (mask rate & mask #segment), each model samples 40 sentences. Samples from different models are randomized in the sample pair. Raters are asked to rank the sentences from the best to the worst in each sample pair. Samples ranking at the $1_{st}$, $2_{nd}$ and $3_{rd}$ place will receive 3, 2, 1 points respectively.

Table 1 reports the perplexity, BLEU scores and human evaluation scores on Yelp dataset when varying mask rate and the number of mask segments. Our model (Self-attn) outwins Seq2Seq and GAN in all three metrics.

**Samples** Table 2 provides an example of the infilling task. The template, ground truth and three samples generated by Self-attn, Seq2Seq and GAN are listed in the table. While Seq2Seq and GAN generate patches that do not fit with the content (line 3: "at appreciated by chinese food"; line 4: "live right of the app"), Self-attn is able to recover the template in a way that is very similar to the ground truth.

### 4.2 PREPOSITION INFILLING

| Mask rate | Mask #segments | Metric | Template | Seq2Seq | GAN | Self-attn |
|---|---|---|---|---|---|---|
| 30% | 1 | BLEU Score | 63.916 | 69.097 | 68.470 | **71.104** |
| | | Perplexity | - | 107.480 | 144.127 | **38.304** |
| | | Human Eval | - | 1.950 | 1.775 | **2.275** |
| | 2 | BLEU Score | 42.233 | 64.174 | 64.337 | **65.914** |
| | | Perplexity | - | 43.044 | 36.704 | **21.028** |
| | | Human Eval | - | 1.838 | 1.975 | **2.188** |
| 40% | 1 | BLEU Score | 56.838 | 61.309 | 61.778 | **63.543** |
| | | Perplexity | - | 202.714 | 230.569 | **44.864** |
| | | Human Eval | - | 2.075 | 1.865 | **2.055** |
| | 2 | BLEU Score | 38.279 | 55.460 | 55.326 | **59.192** |
| | | Perplexity | - | 59.877 | 70.195 | **25.914** |
| | | Human Eval | - | 2.005 | 1.900 | **2.045** |
| 50% | 1 | BLEU Score | 44.369 | 48.865 | 48.861 | **51.55** |
| | | Perplexity | - | 244.862 | 287.415 | **43.688** |
| | | Human Eval | - | 1.725 | 1.863 | **2.412** |
| | 2 | BLEU Score | 32.498 | 42.613 | 42.535 | **44.418** |
| | | Perplexity | - | 99.421 | 107.558 | **32.397** |
| | | Human Eval | - | 1.875 | 1.913 | **2.238** |

Table 1: Quantitative and human evaluations for different mask rates and number of segments.

| | |
|---|---|
| Template | i live __m__ and i was __m__ chinese food . |
| Ground Truth | i live right down the street and i was craving some good chinese food . |
| Seq2Seq | i live at a ten times and i was at appreciated by chinese food . |
| GAN | i live right of the app and i was looking for chinese food . |
| Self-attn | i live in the neighborhood area and i was impressed with the chinese food . |

Table 2: An example from the Yelp data where the template contains two missing portions and 40% of the tokens are masked out.

**Dataset**     For this experiment, we use corpus from *Grimm's Fairy Tale*[1], containing 209 tales collected by the brothers Grimm. We split the long sentences into multiple clauses containing at least 10 but no more than 18 words. The resulting dataset contains 16k sentences for training and 3k sentences for testing with the vocabulary size of 7k.

With the help of the Natural Language Toolkit Bird et al. (2009), we mask out the prepositions(in, at, on...) and articles(a, an, the) in the corpus. Each template contains three blanks and the average mask_rate is 20.9%. Empty masks that remove nothing will be added to the template if there are less than three segments that satisfy such masking rules.

**Quantitative Evaluation**     Since the use-case of prepositions follows a set of rules, and BLEU scores counts matching n-grams in the candidate sentence to n-grams in the reference text, we choose to use BLEU scores to evaluate the generative models' performance for infilling prepositions. Table 3 lists the BLEU score for infilling prepositions on test set. Our model out-performed Seq2Seq and GAN.

**Samples**     Table 4 provides an example of the preposition infilling task. Seq2Seq and GAN are prone to make grammatical mistakes (line 3: "saw at one"; line 4: "the old woman went for, "), which indicates that these two rnn-based generative models failed to grasp the rules of using prepositions. Our model learns the rules and generates prepositions that fit into the template.

---

[1]Grimm's Fairy Tale: `https://www.cs.cmu.edu/~spok/grimmtmp/`

|        | Template | Seq2Seq | GAN | Self-attn |
|--------|----------|---------|------|-----------|
| BLEU   | 38.066   | 70.121  | 73.154 | **80.003** |

Table 3: BLEU scores on the Grimm test set for infilling prepositions

| **Template** | **__m__ old woman went __m__ , but saw __m__ one on the stairs** |
|---|---|
| Ground Truth | the old woman went out , but saw no one on the stairs |
| Seq2Seq | the old woman went with , but saw at one on the stairs |
| GAN | the old woman went for , but saw no one on the stairs |
| Self-attn | the old woman went in , but saw that one on the stairs |

Table 4: An example from the Grimm Tales data where prepositions are masked out.

## 4.3 LONGER CONTENT INFILLING

In this experiment, we test the generative models' performance to fill in longer text contents. We encourage the generative models to generate longer patches by eliminating the information revealed in the template, only a few words are provided as anchors. If all tokens are masked out in the template, then the task reduces to the language model.

**Dataset**     In this dataset, we also train the model on Grimm dataset, sentence number and vocabulary size are the same with section 4.2. We only leave one noun and one verb as anchoring words in the template, and the average mask rate is 81.3%.

**Quantitative and Human Evaluation**     Perplexity is a commonly-used metric in evaluating the performance of language models. The current task may be regarded as a special language model with a few anchoring words, thus we evaluate the generative models' performance with perplexity. We also use human evaluation to judge the quality of the patches. Human evaluation settings are the same with section 4.1. Table 5 reports the perplexity on train set and test data, as well as human evaluation scores. Our model has the smallest perplexity on both training and testing data.

**Samples**     Two examples of long content infilling are provided in Table 9. Even though Seq2Seq tends to generate longer patches, its semantic meanings are weak as it only repeats a few words (line 3: "and the little, and the little,"). GAN is flawed with the same shortcoming and failed to generate meaningful patches. In contrast, our model generates a variety of meaningful patches that fit into the global structure.

## 4.4 MIMIC NEWS GENERATION

Data-driven news generation for automated journalism (Leppänen et al., 2017) is a topic that has attracted much attention. A more challenging task, and also a task with more realistic meanings, is to generate a descriptive sentence from an auxiliary template that contains only a few keywords. We may form such template by keeping the objects or the nouns and removing all the rest.

**Dataset**     We choose to use corpus from NBA scripts provided by Wiseman et al. (2017) to simulate news generation. The dataset consists of sentences containing at least 8 but no more than 16 words. The resulting dataset contains 21k sentences for training and 5k sentences for testing with the vocabulary size of 8k. To mimic the task setting of news generation, we provide a player's name or a team's name as well as a set of numbers in the template. Each template contains three blanks and the average mask rate is 78.1%.

**Quantitative Evaluation**     The task setting for this experiment is similar to the longer content infilling task in section 4.3. We also use perplexity as the evaluation metric. Table 7 lists the perplexity on test set and train set. Our model has the smallest perplexity on both training and testing data.

|  | Seq2Seq | GAN | Self-attn |
|---|---|---|---|
| Test Perplexity | 10.411 | 11.784 | **9.647** |
| Train Perplexity | 9.768 | 11.224 | **8.970** |
| Human Eval | 1.991 | 1.338 | **2.664** |

Table 5: Perplexity on the train/test sets of Grimm Tales for language models with anchor words.

| Template | __m__ sound __m__ be __m__ |
|---|---|
| Ground Truth | if you bear it without letting a sound escape you , i shall be free |
| Seq2Seq | and sound the be and the little , and the little , and the |
| GAN | and sound the be and the , and and |
| Self-attn | the sound said , i will be the king |

Table 6: Examples for language models with anchor words.

**Samples**     Two examples for long content infilling are provided in Table 8. Seq2Seq still suffers from sampling over repeat words (line 3: "defeated the the"). Even though GAN does not generate repeat segments, it fails to learn the script pattern (line 4: "defeated the visiting 114-110"). Our model not only generates meaningful patches, but also provides enough details (line 5: "defeated the Philadelphia_76ers 114-110"), which is suitable for the task of news generation.

|  | Seq2Seq | GAN | Self-attn |
|---|---|---|---|
| Test Perplexity | 10.303 | 7.245 | **6.538** |
| Train Perplexity | 10.675 | 6.857 | **6.738** |

Table 7: Perplexity on the train/test sets of the NBA scripts for language models with anchor words.

| Template | __m__ Toronto_Raptors __m__ 114 - 110 __m__ |
|---|---|
| Ground Truth | The Toronto_Raptors defeated the Detroit_Pistons 114 - 110 on Sunday at the Air Canada |
| Seq2Seq | The Toronto_Raptors defeated the the 114 - 110 on Wednesday at the Center |
| GAN | The Toronto_Raptors defeated the visiting 114 - 110 on Friday . |
| Self-attn | The Toronto_Raptors defeated the Philadelphia_76ers 114 - 110 on Friday . |
| Template | __m__ Bojan __m__ 30 minutes __m__ |
| Ground Truth | Bojan Bogdonavic was not far behind , scoring 22 points in 30 minutes off |
| Seq2Seq | Bojan led the way with with points points 30 minutes , while |
| GAN | Bojan was second on the team , totaling 19 points , 30 minutes , |
| Self-attn | Bojan led the way with 20 points in 30 minutes in the fourth quarter |

Table 8: Examples of the NBA scripts for language models with anchor words.

# 5 CONCLUSION

We have studied the new general task of text infilling, which aims to fill missing portions of a given sentence or paragraph. The task permits an arbitrary number of missing portions each of which can originally have an arbitrary number of tokens. We develop an efficient self-attention model equipped with a segment-aware position encoding mechanism for precise localization and global context modeling. A variety of supervised datasets are created, with varying mask portions and strategies. Our approach significantly improves over common sequence-to-sequence and GAN-based models in the extensive experiments.

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

# A  EXPERIMENTS

| | |
|---|---|
| **Template** | **__m__ sound __m__ be __m__** |
| Ground Truth | if you bear it without letting a sound escape you , i shall be free |
| Seq2Seq | and sound the be and the little , and the little , and the |
| GAN | and sound the be and the , and and |
| Self-attn | the sound said , i will be the king |
| **Template** | **__m__ laid __m__ water __m__** |
| Ground Truth | and when she had finished , she laid it down at the water 's edge . |
| Seq2Seq | and laid the water , and the little , and the little , and the |
| GAN | and laid the water and the , and and the |
| Self-attn | and laid the water in the midst of the forest |

Table 9: Examples for language models with anchor words on Grimm Tales.

| | |
|---|---|
| **Template** | **__m__ Toronto_Raptors __m__ 114 - 110 __m__** |
| Ground Truth | The Toronto_Raptors defeated the Detroit_Pistons 114 - 110 on Sunday at the Air Canada |
| Seq2Seq | The Toronto_Raptors defeated the the 114 - 110 on Wednesday at the Center |
| GAN | The Toronto_Raptors defeated the visiting 114 - 110 on Friday . |
| Self-attn | The Toronto_Raptors defeated the Philadelphia_76ers 114 - 110 on Friday . |
| **Template** | **__m__ Bojan __m__ 30 minutes __m__** |
| Ground Truth | Bojan Bogdonavic was not far behind , scoring 22 points in 30 minutes off |
| Seq2Seq | Bojan led the way with with points points 30 minutes , while |
| GAN | Bojan was second on the team , totaling 19 points , 30 minutes , |
| Self-attn | Bojan led the way with 20 points in 30 minutes in the fourth quarter |

Table 10: Examples of the NBA scripts for language models with anchor words.

