# OpenReview forum: "Text Infilling"
_ICLR.cc/2019/Conference_

### Official Review · AnonReviewer1 · 2018-10-24
**Lack of novelty**

**Rating:** 6
**Confidence:** 4

**Review:**

Pros:
This paper targets an interesting and important task, i.e. general text filling, where the incomplete sentence can have arbitrary number of blanks and each blank can have arbitrary number of missing words.  Convincing experiments are conducted both qualitatively and quantitatively.

Cons:
1. Lack of novelty in the proposed method. Specifically, such impression mainly comes from the writing in Sec 3.2. When discussing details of the proposed method, authors keep referring to [A], indicating heavily that authors are applying an existing method, i.e. yet another application of [A].  This limits the novelty. On the other hand, this also limits the readability for anyone that not familiar with [A]. Moreover, this prevents authors discuss the motivation behinds their architectural choices. Whether [A] is the optimal choice for this task, and can there be alternative options for its components. For example,  could we use a rnn to encode x_template_i and use the encoding as a condition to fill  s_i?

[A] Vaswani, Ashish, et al. "Attention is all you need." Advances in Neural Information Processing Systems. 2017.

2. Discussion in Sec 3.2 and Figure 2 are not clear enough. I didn't get a picture of how the proposed method interacts with requirements of the task. For example, in sec 3.2.3, what represent queries, keys and values respectively are unclear. And authors mention "template-decoder attention" layers, where  I didn't find in Figure 2.

3. Is not very straightforward that how the baselines Seq2Seq and GAN are applied to this task, where necessary information is missed in the experiment section.

---

### Official Review · AnonReviewer2 · 2018-11-01
**questions about experiments**

**Rating:** 5
**Confidence:** 4

**Review:**

Summary
This paper proposes an approach to fill in gaps in sentences. While usually generation assumes a provided left context, this paper generalizes generation to any missing gap of any number of words.
The method is based on Transformer which does attention on all the available context and generates in a left to right manner.
The authors validate their approach on rather small scale datasets. I am unclear about several details of their empirical validation.

Relevance: this work is on text generation, which is very relevant to this conference.

Clarity: the description of the method is very clear, less so the description of the empirical validation.

Novelty: The method per se is not novel, but the combination of method and application is.

Empirical validation: The empirical validation is not very clear and potentially weak.
First, I did not understand the baselines the authors considered: seq2seq and GAN. There are LOTS of variants of these methods and the references cited by the authors do not immediately apply to filling-in tasks. The authors should explain in detail what these baselines are and how they work.
Second, MaskGAN ICLR 2018 was proposed to solve the filling task and code is publicly available by the authors of that paper. A direct comparison to MaskGAN seems a must.
Third, there are lots of details that are unclear, such as : how are segments defined in practice? the ranking metric is not clearly defined (pair-wise ranking or ranking of N possible completion?). Could the authors be a little bit more formal?
Speaking of metrics, why don't the authors considered precision at K for the evaluation of small gaps?
Fourth, the datasets considered, in particular the Grimm's dataset, is very small. There are only 16K training sentences and the vocabulary size is only 7K. How big is the model? How comes it does not overfit to such a small dataset? Did the authors measure overlap between training and test sets?

More general comments
The beauty of the proposed approach is its simplicity. However, the proposed model feels not satisfactory as generation proceeds left to right, while the rightmost and the leftmost missing word in the gap should be treated as equal citizens.

Minor note: positional embeddings are useful also with convolutional models.

---

### Official Review · AnonReviewer3 · 2018-11-02
**The authors present an interesting new conditional generation task, but the paper lacks critical implementation details and the experimental setting is too restricted to fully support the claims.**

**Rating:** 3
**Confidence:** 4

**Review:**

The paper proposes a setting for evaluation of a text infilling task, where a system needs to fill in the blanks in a provided incomplete sentences. The authors select sentences from three different sources, Yahoo Reviews, fairy tales, and NBA scripts, and blank out words with varying strategies, ranging from taking out prepositions and articles to removing all but two anchor words from a sentence. On this data, they compare the performances of a GAN model, Recurrent Seq2seq model with attention, and Transformer model in terms of BLEU, perplexity, and human evaluation.

The setting is certainly interesting, and the various data creation strategies are reasonable, but the paper suffers from two main flaws. First, the size of the data set is far from sufficient. Unless the authors are trying to show that the transformer is more data-efficient (which is doubtful), the dataset needs to be much larger than the 1M token it appears to be now. The size of the vocabularies is also far from being representative of any real world setting.

More important however is the fact that the authors fail to describe there baseline systems in any details. What are the discriminator and generator used in the GAN? What kind of RNN is used in Seq2seq? What size? Why not use a transformer seq2seq? How exactly is the data fed in / how does the model know which blank it's generating? It would be absolutely impossible for anyone to reproduce the results presented in the paper.

There are some other problems with the presentation, including the fact that contrary to what is suggested in the introduction, the model seems to have access to the ground truth size of the blank (since positional encodings are given), making it all but useless in a real world application setting, but it is really difficult to evaluate the proposed task and the authors' conclusions without a much more detailed description of the experimental setting.

---

### Meta-Review · Area_Chair1 · 2018-12-14
**rejection**

**Confidence:** 4
**Recommendation:** Reject

**Metareview:**

although the problem of text infilling itself is interesting, all the reviewers were not certain about the extent of experiments and how they shed light on whether, how and why the proposed approach is better than existing approaches.